# Influence of Social Support and Subjective Well-Being on the Perceived Overall Health of the Elderly

**DOI:** 10.3390/ijerph18105438

**Published:** 2021-05-19

**Authors:** Valeria Farriol-Baroni, Lorena González-García, Aina Luque-García, Silvia Postigo-Zegarra, Sergio Pérez-Ruiz

**Affiliations:** Departament of Psychology, Faculty of Health Sciences, Universidad Europea de Valencia, 46010 Valencia, Spain; valeriaadriana.farriol@universidadeuropea.es (V.F.-B.); ainarosa.luque@universidadeuropea.es (A.L.-G.); silvia.postigo@universidadeuropea.es (S.P.-Z.); sergio.perez@universidadeuropea.es (S.P.-R.)

**Keywords:** elder people, social support, subjective well-being, perceived overall health

## Abstract

Scientific interest in the positive aspects of aging and the development of healthy aging has increased, given the need to ensure older people well-being and quality of life. In this sense, social support and some sociodemographic variables may have a not yet entirely clear role. The main objective of this work was to analyze the predictive relationships of age, marital status, social support and subjective well-being on the general perception of the health of a group of elderly people. The participants were 137 people (77.4% women) between 61 and 91 years old (*M* = 73.11; *SD* = 6.22); 56.9% of them had a partner and 40.1% did not. The path analysis tested indicates that social support has an indirect predictive value on perceived overall health through its influence on subjective well-being. Age and life satisfaction are the most important direct predictors of perceived overall health. Conclusions highlight the need to delve into the study of explanatory factors of the general perception of the health of the elderly and promote interventions to facilitate the development of an appropriate social support network and increase the subjective well-being of this group.

## 1. Introduction

People’s life expectancy has increased throughout the world, which has led to growth in the presence of older people in society. For example, in Spain, the data indicate that at the beginning of the 20th century, the percentage of older people was 5.2%. In 2019, this percentage increased up to 19.4%; for 2033, the population over 65 years will be 25.2% and 38.7% in 2050 [1,2,3,4]. This worldwide situation has led to the growing scientific interest in the maintenance and development of well-being and quality of life at this stage of life.

The research on the elderly has mainly focused on the negative view of old age, in which the multiple losses that occur, and the pathologies associated with aging, were highlighted [5]. However, to contribute to the study of the promotion of well-being within the elderly, it is essential to focus on the study of deterioration and understanding the facilitating aspects of healthy aging [6]. In this line, previous research has analyzed the relationship between age, the family situation [7], social participation [8], and the elderly well-being and quality of life, finding discrepant results about the importance of some variables [9,10]. Along these lines, this study aims to contribute to the existing body of literature about the promotion of well-being in old age and analyze a current sample of participants to delve into the relationship between the social support perceived by older people, indicators of well-being (i.e., satisfaction with life and self-esteem), and perceived general health.

Centering in the construct of quality of life, it is understood as a subjective, global, and multidimensional perception that considers the individual, the environment surrounding him, and the relationships between the different variables that seem to affect this perception [11]. Among other indicators of well-being, satisfaction with life and happiness significantly influences the quality of life [12], and other variables such as social relationships, the environment where the elderly live, or their wealth [13]. Self-reported health or perceived overall health has been introduced in the scientific literature as a general indicator of the quality of life [14]. The perception of overall health considers the current physical state of the person and the level of emotional adjustment and seems to correlate with life satisfaction, self-esteem, and emotional symptoms such as depression [15,16]. In addition, a good perception of health is associated with lower mortality and a reduction in the use of health resources [17]. Under this information, in this work, we use the global perceived health indicator in older people and analyze its relationship with some psychological constructs of well-being to understand which variables need to be used to promote optimal aging [13].

Focusing on the elderly experiences of well-being, the two indicators used in the present study are satisfaction with life and self-esteem. Life satisfaction is defined as the person’s assessment of their actual shape compared to another experienced moment or other people [18,19]. Life satisfaction is based on the greater experimentation of positive than negative emotions, and it is derived in turn from participation in meaningful activities [20]. Previous literature has defended the importance of life satisfaction, perceived health, social support, and coping styles to the subjective well-being of older adults [21].

On the other hand, self-esteem is another widely used indicator to evaluate well-being in the scientific literature, and it is defined as the global assessment that the person makes of himself, including elements of satisfaction and affection [22]. Self-esteem also seems to be a good predictor of quality of life, and it is considered a protective factor of healthy aging [17,23,24]. In addition, it has been consistently observed that there is a positive relationship between satisfaction with life and self-esteem [25].

A factor that can influence the quality of life perceived overall health, and well-being of the elderly is the perception of social support, which is understood as the subjective evaluation that an individual makes about the adequacy of the social support received, that is, the subject’s appreciation of the quality of their significant relationships and the support available from them [26]. Some authors have highlighted the importance of the elderly perceiving having adequate support networks to promote their experiences of emotional stability or the feeling of being cared for by others [27], which has been related to subjective well-being in old age [7,28]. Therefore, establishing a dense network of contacts and alliances can help face the old age challenges [29]. However, some studies have observed that as age increases, social support decreases, and older people, particularly those who live alone, constitute one of the most vulnerable groups that refer to feelings of loneliness [30].

Some sociodemographic factors linked to social support can influence the quality of life of older people. For example, it has been observed that marital status can positively influence the well-being of the elderly since the partner can be important support [31]. On the other hand, age influences the perception of well-being [32], and it has been associated with a worse perception of health, more significant physical deterioration, a greater probability of suffering from a chronic disease, and living alone [33].

In addition to the psychosocial factors explained so far, additional factors have been shown to impact the quality of life of the elderly. One of these main factors is physical activity, specifically group physical activity, which has been shown to benefit the elderly quality of life, general health, social function, and life satisfaction [32]. Educational interventions to promote health have also been recognized to favour well-being and quality of life in old age [33].

Considering state of the art, the main objective of this study is to analyze the mentioned relationships in a path analysis that allows analyzing the possible influence of social support perceived by older people on their well-being (using two indicators: satisfaction with life and self-esteem), and of these in the perceived general health. These variables have shown some importance as protective factors for healthy aging; however, their relationships are not evident, especially considering sociodemographic factors such as marital status and age. In order to answer this question, the following specific objectives were proposed: (1) to analyze the relationships between age, marital status, perceived social support, satisfaction with life, self-esteem, and perceived general health; and (2) analyze the predictive relationships of these variables on the perceived general health through path analysis that allow all relationships to be analyzed together.

## 2. Materials and Methods

### 2.1. Participants

People older than 60 years with no severe illness were invited to participate in the study. The study’s final sample was composed of 137 participants (women: *n* = 106; 77.4%). Further details about the sample are shown in the results section.

### 2.2. Instruments

Sociodemographic data. Sociodemographic information was obtained through an ad hoc questionnaire with questions about the age, sex, place of residence, country of birth, and marital status of the people who participated in the study.

#### 2.2.1. Perceived Social Support

The Duke-UNC-11 Questionnaire [34] in its Spanish version was used to evaluate the social support perceived by the elderly in daily life [35]. This instrument is made up of 11 items divided into two subscales: (1) the confidential support subscale, with 7 items such as “I can to talk to someone about my problems at work or home”, refers to the perception of having people who can help in difficult situations; and (2) the affective subscale, with 4 items such as “I have people who care about what happens to me”, which refers to the perceived facilities for social relationships and emotional communication. The response scale is a 5-point Likert type, where 1 refers to much less than you want and 5 is as much as you want.

#### 2.2.2. Satisfaction with Life

The Satisfaction with Life Scale [18] was used in its Spanish version [36]. It is composed of a single dimension evaluated by 5 items: “So far, I have gotten from life the things that I consider important”. The response scale is a 7-point Likert-type, where 1 is strongly disagree, and 7 is strongly agree. The validity and reliability of the scale have been confirmed in previous studies with an adult population [36].

#### 2.2.3. Self-Esteem

The Spanish version [37] of the Rosenberg Self-Esteem Scale [21] was used to evaluate this variable, which is unidimensional and consists of 10 items with a 4-point Likert-type response scale, where 1 is strongly disagree, and 4 is strongly agree. The scale presents 5 items with positive statements such as “I am capable of doing things as well as others” and 5 inverse items: “I would like to have more respect for myself”. The reliability and validity of the scale have been previously evidenced in the elderly population [38].

#### 2.2.4. Perceived Overall Health

This variable has been evaluated using the Spanish version [39] of the Visual Analogue Scale of the EuroQol-5D Questionnaire [40], which is composed of a single item to which the participants must answer with a number in units ranging between the minimum value of 0 (representing “the worst imaginable state of health”) and the maximum value of 100 (representing “the best imaginable state of health”).

### 2.3. Procedure

This study presents a cross-sectional empirical–observational design with convenience sampling. Before starting the study, the approval of the ethical committee of the European University was obtained. The requirements to participate in the study were: (1) be over 60 years old, (2) live autonomously (without dependency conditions), and (3) have no disabling illness. Before proceeding with the data collection, the study´s objective was explained to the participants, and we collected their signed informed consent to participate in the research. Members of the research team helped the participants and answered questions during the completion of the questionnaire, which took approximately 30 min.

### 2.4. Data Analysis

First, the reliability of the measurement instruments used was analyzed using Cronbach’s alpha coefficient, and the main variables were described. For the relationship between variables, bivariate correlations (Pearson’s coefficient) were performed using the statistical software SPSS Statistics V23 (IBM Corp., Armonk, NY, USA). For the multivariate analysis, path analysis was used to explain the general perception of health from all the independent variables simultaneously, considering the possible predictive relationships and correlations between them. In the path analysis, the Maximum Likelihood technique was used through the EQS 6.1 statistical program (Multivariate Software, Inc., Encino, CA, USA) [41]. This technique allows a minimal deviation of the data for normality, which was fulfilled in the present study, since Mardia’s normalized estimate was 2.5, which is below the recommended cut-off criterion of 3. However, in addition to the indices of adjustment CFI (Comparative fit index) and GFI (Goodness of fit index), robust indices of these estimators were also used. In addition, other fit indicators are reported, such as the standardized square means of the residual square root (SRMR, which must approach or be less than 0.08 to be indicative of a good fit of the model to the data). The root mean square error of approximation (RMSEA, which represents the closeness of the fit with values lower than 0.08) and the normalized chi-square statistic (which allows detecting oversized models with values lower than 1) [42].

## 3. Results

The participants of the study were 137 people (women: *n* = 106; 77.4%) aged between 61 and 91 years (*M* = 73.11; *SD* = 6.22); residents in 11 different locations in the province of Valencia, Spain. Regarding marital status, 78 participants had a partner (58.6%), 55 did not have a partner (41.4%), and 4 did not provide information in this regard.

The results of the study show that all the questionnaires presented adequate fit indices (Table 1): social support questionnaire (*α* = 0.82), confidential support subscale (*α* = 0.78), affective support subscale (*α* = 0.70), life satisfaction scale (*α* = 0.81), and self-esteem scale (*α* = 0.71). The means of the study variables indicate that the participants present a good level of social support, self-esteem, and satisfaction with life. The mean of perceived health is 67.04 (*SD* = 18.94, range = 0–100).

The correlational analyses (Table 2) indicate that the general perceived health significantly correlates with age (*r* = −0.23) and the indicators of well-being: satisfaction with life (*r* = 0.22) and self-esteem (*r* = 0.19). In turn, these last indicators significantly correlate with each other (*r* = 0.19) and with social support (except self-esteem and the affective support subscale). Finally, social support significantly correlates with age (*r* = 0.23), indicating that older people perceive greater social support in the two measured dimensions: confidential (*r* = 0.19) and affective (*r* = 0.26).

Results of the path analysis are shown in Figure 1. This model showed a good fit of the data: the chi-square coefficient was not statistically significant and greater than 3 (*χ2* = 9.795, *p* = 0.20), the CFI and GFI indices were greater than 0.9 (CFI = 0.95, GFI = 0.97), and the errors were less than 0.06 (SRMR = 0.06, RMSEA = 0.05). The results of the model indicate that age positively and significantly predicts social support and negatively the general health perceived by the elderly. In turn, social support positively and significantly predicts the perceived health and self-esteem of the elderly, and the latter positively and significantly predicts the participants’ perception of health. The only relationships tested in the model that are not significant are the prediction of social support by the marital status (dichotomized to partner or no partner) and predicting the perceived general health by self-esteem. The variance explained by the model is not very high (R^2^ = 0.11), but there is a direct predictive capacity of age (SEB = −0.23, *p* = 0.001) and satisfaction with life (SEB = 0.20, *p* = 0.001) on the general health perceived by the elderly. Social support would have an indirect predictive value on perceived general health through its influence on life satisfaction (SEB = −0.34, *p* = 0.001).

## 4. Discussion

The main objective of this work was to analyze the relationships between two sociodemographic variables, social support, two classic indicators of subjective well-being, and the perceived general health of the elderly, mainly centering on the prediction of the general perceived health.

Regarding the sociodemographic variables, the results indicate that age decreases the perceived general health directly, which is consistent with a natural deterioration of health in old age, and indirectly through less social support. In the present study, it has been observed that age positively predicts the perception of social support, which in turn predicts the experiences of well-being and the general perception of the health of the elderly. However, previous literature has shown that old age reduces the social support network available [28,31,43]. Future studies need to analyze these relationships to understand better the factors that may interfere with their relationship.

The marital status does not seem to influence the perceived general health, either directly or indirectly. These last data are controversial since previous studies have observed a relationship between having a partner and perceiving greater social support [29,30], which would result in a higher quality of life. This discrepancy with the previous literature may be a statistical phenomenon due to the sample size, since the correlations between marital status and social support are at the limit of significance, and it is possible that, with a larger sample, the previous results were corroborated. Likewise, having a partner and living with other people who can also provide confidential and emotional support may be another variable affecting social support.

The social support perceived by people over 60 years of age has an indirect influence on their perceived general health through its effect on subjective well-being. Thus, the social support perceived by the elderly who participated in the study significantly contributes to their experiences of life satisfaction and their self-esteem. These results are consistent with previous studies [39,44,45]. These relationships suggest the importance of promoting both adequate social support and the subjective well-being of the elderly since both variables are closely related and contribute to their perceived overall health. Distinguishing between confidential support, referring to the availability of people who can help, and affective support, referring to the satisfaction of emotional needs, the former seems to be more influential in well-being, especially in life satisfaction. In this line, it seems that elderly emotional needs may not be decisive for their perception of well-being and overall health. Alternatively, confidential support fulfils the function of support in problems and conflicts, but it indirectly helps satisfy emotional needs through the feeling of being understood.

Finally, the present study corroborates the role of subjective well-being in the perception of the general health of the elderly, showing how satisfaction with life experienced by the elderly positively predicts their general health. Previous studies in elderly people have presented similar results, showing that when older persons feel satisfied with their lives, they are more likely to feel high self-esteem and good perceived general health [11,31,39,46]. In the present study, self-esteem has not turned out to be a positive predictor of the perception of health, so a greater analysis of these variables is required in future studies.

Before concluding, it should be noted that most of the participants in this study are women, so we must be cautious when generalizing the results to the general population of older people. In addition, all the participants were contacted through associations for the elderly (e.g., retirement homes, associations for the elderly, or adult schools), so these people may perceive greater social support and have greater subjective well-being and perceived general health than people who do not go to these centers and who have not been able to access. Future studies in this field must incorporate other factors such as physical activity or educational programs and the examination of age-friendly environments to understand better the factors that contribute to elderly well-being and quality of life [32,33].

## 5. Conclusions

The path analysis tested in this study explains 11% of the variance in the perceived general health of the elderly, showing the beneficial effects of social support on perceived health in the elderly through its contribution to better subjective well-being (mainly through perceived satisfaction with life). These results are consistent with the previous literature [30]. They allow us to clarify some aspects, such as the scarce role of marital status in social support and the indirect influence of social support on the quality of life when all these relationships are considered together. The results suggest the importance of ensuring that the elderly perceive good social support, as this predicts greater satisfaction with life, contributing significantly to their overall perception of health. It is necessary to continue investigating the role of self-esteem in this population group since although our results do not show a direct effect of self-esteem on general health, it does show positive relationships with social support and life satisfaction in other studies [47].

Future research could advance in this area of study by expanding the sample and including greater heterogeneity in the sex of the participants. Research on aging from a sex perspective is still scarce [48], and it would be interesting to test sex differences in old age, as some studies have shown differences in the perceived quality of life of the elderly [22]. In addition, given the variability of sociodemographic aspects that make up a social support network, it would be necessary to delve into other sociodemographic aspects and the social relationships of the elderly [49], taking into account the influence of age, as well as the structure, functionality and quality of the relationships, since, although the partner is a good source of support, it may not be the only one. Moreover, new factors such as the environment need to be explored within these to understand the elderly quality of life better. Finally, we believe that it is necessary to continue developing research to promote the personal and social resources of the elderly group to promote healthy aging.

## Figures and Tables

**Figure 1 ijerph-18-05438-f001:**
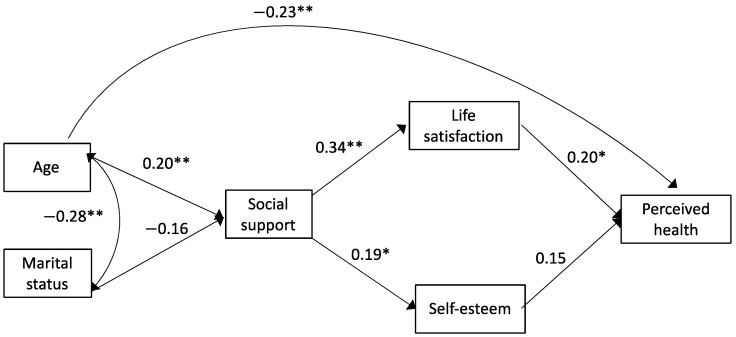
Path analysis to explain perceived general health. Note: Errors are not displayed to improve the visibility of the model. The estimates of the parameters (standardized beta coefficient for regression analyzes, indicated with one-way arrows; and Pearson’s coefficient for correlation, indicated with a two-way arrow) that were statistically significant are indicated: * *p* ≤ 0.05; ** *p* ≤ 0.01.

**Table 1 ijerph-18-05438-t001:** Descriptive statistics of the variables used in the study.

	*n*	Range	*M*	*SD*	Cronbach’s Alpha
Social support	137	1–5	3.93	0.74	0.82
Confidential	137	1–5	3.78	0.82	0.78
Affective	137	1–5	4.20	0.79	0.70
Life satisfaction	137	1–7	4.99	1.27	0.81
Self-esteem	137	1–5	3.14	0.50	0.71
Perceived general health	137	1–100	67.04	18.94	-

**Table 2 ijerph-18-05438-t002:** Correlations between the study variables.

	1	2	3	4	5	6	7	8
Age	-							
2. Marital status	0.35 **	-						
3. Social support	0.23 **	−0.17	-					
4. SS confidential	0.19 *	−0.16	0.95 **	-				
5. SS affective	0.26 **	−0.16	0.83 **	0.62 **	-			
6. Life satisfaction	0.04	0.06	0.34 **	0.34 **	0.24 **	-		
7. Self-esteem	−0.08	0.12	0.21 *	0.23 **	0.11	0.19 *	-	
8. General health	−0.23 **	0.09	0.08	0.12	−0.01	0.22 *	0.19 *	-

Note: Marital status was coded in two categories: 1 = without a partner (single, widowed) and 2 = with a partner or married. * *p* < 0.05; ** *p* < 0.001.

## Data Availability

Data are available from the authors. Any interest please contact the author of correspondence.

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
