# Peer review of "Influence of Social Support and Subjective Well-Being on the Perceived Overall Health of the Elderly"

_ijerph, 2021, doi:10.3390/ijerph18105438_

Round 1

Reviewer 1 Report

This is a manuscript about how some social factors influence quality of life in a limited sample of Spanish individuals.

  • One of the main concerns with this manuscript is that authors do not justify or argument in favor why their study is important. They should acknowledge the fact that this objective has been already addressed a number of times and in a bigger sample. Even for Spanish individuals, a lot has been written on this matter with the SHARE study. So, it seems that the authors need to make a better effort into presenting the state of the art on this topic and then make a better case out of their work in order to highlight the relevance of it.
  • The introduction fails to integrate the concepts that are to be explored in the study, needs a better arrangement of the ideas in order to them to converge into the objective of the study. In their current form, these are scattered topics that not seem to be related. 
  • There are some statements that need to be supported by references. For example, authors state that '...subjective well-being, the most robust indicator is satisfaction with life...' Please add a reference that truly reflects this assertion, based on facts not other opinions. Check the rest of the manuscript because it has a number of this unsupported assertions all along. 
  • References could use an update. 
  • Please do not include results into the methods section. The description of the population may suit better into the results and not in the first lines of the methods section. Moreover, the description of the origin of the individuals seems out of place. 
  • Details on the procedures are a bit too long, in particular the details on the informed consent.
  • It is not clear why showing the results of the consistency with Cronbach´s alpha and the correlations, were important. A better way to present the results and more informative to the reader, would have been to describe them by sex or groups of age. 
  • Presented results are not surprising, and as previously noted, this has been widely explored in the literature. 
  • Please avoid prejudicial statements such as "...which is consistent with a natural deterioration of health in old age..."
  • The rest of the discussion lacks of a clear argument on what the results presented actually mean in the context of what is already known. 

Author Response

We thank the reviewer by the extensive revision of the manuscript. Please see the attachment to read the answers.

Reviewer 2 Report

 The main objective of this work was to analyze the predictive relationships of age, marital status, social support and subjective well-being on the quality of life of elderly people. In general, the text is written in a clear and concise manner, with a pertinent introduction and a discussion that summarizes and comments on the results found.

These are the  aspects that should be reviewed in the same order of appearance of the manuscript are detailed below:

  • It is not appropriate to use a single perceived general health item to assess quality of life. As stated in page 2: "Quality of life is understood as a subjective, global and multidimensional perception that takes into account the individual, the environment that surrounds him and the relationships between the different variables that seem to affect this perception". The arguments provided on page 3 are not enough, since with the item used you can know the physical state of health but not infer the level of emotional adjustment. Authors should not refer in the text to quality of life but to perceived overall health (which is what they have measured and not quality of life).
  • It is not clear which procedure was used to analyze the data: on page 2 the term “structural regression model” is used, and on page 4 the generic term Structural Equation Model is used. If, as I believe, the test scores have been used in the model the correct term is Path Analysis and the variables should be drawn with rectangles in the model, since they are manifest variables. If you have used a model with latent variables, Figure 1 is incomplete.
  • The format of Table 1 should be improved. The names of the variables are not in line with the rest of the elements of the table.
  • The format of Table 2 should be improved, expanding the width of the first column, where the variable names are located.
  • The authors said that “age positively and significantly predicts social support and the general health perceived by the elderly” (pag 5, line 189). However, the relationship between age and perceived health is negative.
  • The variable “Marital Status” should be eliminated from the SEM model, since it has no significant relationship with any of the variables in the model (see table 2).
  • It is necessary to carry out a mediation analysis, and for that it is not enough to provide the indirect effect data. To prove that mediation exists, it is necessary to use a specific procedure, and verify that the simple relationship between X and Y [c] shrink or disappear when the mediator is present in the model (ie, is c΄ <c, or is c΄ = 0).

Author Response

We thank the reviewer for the extensive comments on the manuscript. Please see the attachments to read our answers. 

Reviewer 3 Report

This study examined how age, marital status, social support, life satisfaction and self-esteem predicts quality of life in a sample of 137 older adults in Spain using cross-sectional data and structural equation modeling. Perceived general health using the single-item of the VAS of the EuroQol-5D was used as a proxy for quality of life. While the research question is of longstanding importance in the field of aging, I think the methods would require major improvements before this paper can be published. General comments and suggestions are as follows.

General comments:

1. As only self-report data were examined, it is clear that all of the proposed predictors were subjective perception of the constructs. It was unclear if the authors intended to highlight the differential roles of subjective vs objective indicators of the constructs (e.g. perceived social support vs. number of people in social network). If so, the objective indicators need to be included.

2. Psychosocial factors aside, additional factors (e.g. SES, physical mobility, an age friendly environment) likely play important roles in affecting the quality of life of older adults. However, they are not addressed in the study conceptually or empirically.  The literature review and the empirical study as presented in this paper need to be significantly expanded to acknowledge, if not empirically examine as well, the roles of these other factors.  

Comments on the methods:

  1. The sample size is too small for a SEM analysis and therefore the study lacks sufficient statistical power for the proposed analyses. The authors may consider increasing the sample size.
  2. When using SEM, rather than looking at the fit of the data to 1 proposed conceptual model, it would be more appropriate to compare how well the data fit several proposed conceptual models. The authors only examined 1 model, rendering the results difficult to interpret. Taking another step back, given the current sample size and the number of predictor variables, it would be more appropriate to use linear regression analyses or path analyses as opposed to SEM to examine the proposed conceptual model.

The authors are suggested to address these issues to strengthen the methodology of this paper.

Overall, given the question of how psychosocial factors play a role to contribute to quality of life is of longstanding significance in the field of aging, it is imperative that this authors demonstrate how this study makes novel or innovative contributions in addressing the question, and how the data and the methods enable the study to do so.   The study requires major improvement to achieve these goals. 

Author Response

(The authors gave the same response as above.)

Round 2

Reviewer 1 Report

None.

Reviewer 2 Report

The authors have made a commendable effort to resolve the concerns indicated in the review. Regarding the suggestion to include a specific analysis of mediation, it is true that Path analysis is a good framework to study it, but it requires specific analyzes. The most common is by bootstrapping. In any case, given that in the text the results of the model are only interpreted in terms of direct and indirect effects and the term mediation does not appear, it is not considered essential to carry out the mediation analysis.

Reviewer 3 Report

The authors have addressed my comments satisfactorily in this revision.